# ALIEN: Assisted Learning Invasive Encroachment Neutralization for Secured Drone Transportation System [note 1]

**DOI:** 10.3390/s23031233

**Published:** 2023-01-20

**Authors:** Simeon Okechukwu Ajakwe, Vivian Ukamaka Ihekoronye, Dong-Seong Kim, Jae-Min Lee

**Affiliations:** Department of IT Convergence Engineering, Kumoh National Institute of Technology, Gyeongbuk 39253, Republic of Korea

**Keywords:** assisted learning, deep learning, detection, drone transportation, invasion, real-time, security, surveillance

## Abstract

Priority-based logistics and the polarization of drones in civil aviation will cause an extraordinary disturbance in the ecosystem of future airborne intelligent transportation networks. A dynamic invention needs dynamic sophistication for sustainability and security to prevent abusive use. Trustworthy and dependable designs can provide accurate risk assessment of autonomous aerial vehicles. Using deep neural networks and related technologies, this study proposes an artificial intelligence (AI) collaborative surveillance strategy for identifying, verifying, validating, and responding to malicious use of drones in a drone transportation network. The dataset for simulation consists of 3600 samples of 9 distinct conveyed objects and 7200 samples of the visioDECT dataset obtained from 6 different drone types flown under 3 different climatic circumstances (evening, cloudy, and sunny) at different locations, altitudes, and distance. The ALIEN model clearly demonstrates high rationality across all metrics, with an F1-score of 99.8%, efficiency with the lowest noise/error value of 0.037, throughput of 16.4 Gbps, latency of 0.021, and reliability of 99.9% better than other SOTA models, making it a suitable, proactive, and real-time avionic vehicular technology enabler for sustainable and secured DTS.

## 1. Introduction

Regardless of the plethora of potential advantages that prompted the creation of unmanned aerial vehicles (UAV), the proliferation of drones is likely to cause unprecedented disruption of future air-based transport ecosystems due to their susceptibility to societal and cybersecurity threats from non-state actors [1,2]. The author in [3] argues that security and privacy issues are frequently raised in relation to novel concepts propelled by innovation. Ironically, the more sophisticated an innovation becomes, the greater the need for introspective transparency in the enabling technology’s decision-making process [4] to ensure security. The viability and acceptability of innovation depend on the deployment of reliable methodologies and designs to address security concerns. A drone transportation system (DTS) is an emerging mobile cyber–physical system (CPS) that comprises the convergence of real-time control systems, distributed systems, embedded systems, and edge networks (wireless sensor networks) for the smart mobility of goods and services.

In recent years, unmanned aerial vehicles (UAV) and other intelligent autonomous systems (IAS) have become increasingly common, especially for “priority-based logistics,” which has raised concerns about the reliability of the technologies underpinning these systems and, consequently, the value placed on human lives [5,6]. As a result, if creative, adaptive strategies are not in place to prevent the misuse of UAV technology, an expansion in the intelligence and autonomy of UAVs will jeopardize their future use [7,8]. According to the authors in [9], a drone invasion is a calculated and planned attempt to sabotage operations, confuse workers, destroy installations, and perhaps leak important information. Such invasions are carried out through unauthorized access to restricted areas. According to the United States Federal Aviation Authority (FAA), there were 8344 drone-related violations between 2016 and 2019 [10] despite available counter-invasive technologies. Furthermore, there are already instances of invasions employing quad-copter UAVs (such as the DJI Phantom) for recreational, logistics, and consumer use to destroy industrial sites, transport dangerous goods over borders, and assault important installations [11,12]. Therefore, if sufficient proactive surveillance systems are not in place to confirm a drone’s legitimacy and appropriateness, the sight of a UAV deployed for logistics in a smart city over a drone transportation network (DTN) can overwhelm a civilian area, inflict distress, and cause severe emotional torture. Consequently, this chaotic scenario might endanger the viability of DTS as an intelligent vehicle transportation enabler.

The neutralization of an invasive drone encroachment in a DTS is a cooperative, multivariate, and cognitive decision-making process that depends on the quality of the data. Neutralization activities include deciding whether a drone is legitimate, determining its jurisdiction, and determining its integrity within the airspace transportation network. Sadly, there are not many trustworthy datasets available in this field. Furthermore, prior research focused on single-modal, license/authorization, flight path, or operating boundary drone detection, paying little to no attention to the attached objects, rendering its decision-making procedures ineffective [13,14,15]. Didactically, according to the authors in [5,6], a drone’s perceived risk in a DTN depends on how well one understands the source, the attached objects, and the drone’s network behavior. Additionally, due to its inherent shortcomings, using a single detection method to accomplish this purpose is ineffective. Moreover, the majority of neutralization techniques are either manual, militaristic, or reactive [10,16,17]. Undoubtedly, improper neutralization of invasive drones in a DTN, incorrect visual recognition of conveyed objects, and late or inaccurate detection can harm the future of DTS. Hence, a trustworthy tracking method should not only notify users that something is present in the network (detection) but also give a detailed description of the predetermined characteristics it is using (identification).

Therefore, to address various drone-related threat dynamics in a timely, accurate, efficient, and situation-aware manner, sustainable DTS requires a synergistic, scalable, and multifaceted networked-integration surveillance approach that makes use of 5G innovations and artificial intelligence (AI) capabilities [18]. This study is a novel attempt to address these problems by proposing a collaborative approach for determining a flown drone’s legitimacy using a fusion of a vision-based deep learning (DL) model and lidar technology. Assisted learning (AL) is an emerging machine learning framework that aims at autonomy, model privacy, data privacy, and unlimited access to local resources [19]. Unlike federated learning, the goal of AL is to provide protocols that significantly expand the learning capabilities of decentralized agents by assisting each other with their private modeling processes without sharing confidential information. Using neural networks and related technologies, assisted collaborative cognition in this context entails comprehending the activity and behavior of the drone, identifying its origin, and remembering its established relationships within a transportation network or route before choosing the best course of action among various dynamic scenarios from different network-based detection sources.

The specific contributions of this paper are:To design a multimodal invasive drone detection network that can detect and classify various drones in a DTN operating in ambient environments, estimate their range, and identify the conveyed objects’ characterization in real-time.To present a reliable and robust dataset for trustworthy and real-time malicious drones with attached object detection and elicitation that encompasses most real-life scenarios in a DTN.To develop a mutivariate perceived danger analysis in tandem with the scenario’s current characterization required for cognitive data-centric decision making.To formulate a multivariate situation-aware encroachment neutralization strategy for ascertaining the appropriate decision for any given dynamic state.To evaluate the model’s performance with other state-of-the-art (SOTA) models and approaches.

This paper is organized into the following sections: Section 2 provides a brief overview of existing DTS surveillance methods. Section 3 presents the proposed system design, model architecture, and methodology. Section 4 highlights the experimental results, discussion, and performance evaluation, and Section 5 concludes with a promising research direction and open issues.

## 2. Overview of Invasive Drone Encroachment Techniques

This section provides an accessible overview of several research initiatives to stop recurrent intrusions and disturbances in a DTN using convergence approaches, as well as investigations into drone surveillance tactics in DTS.

### 2.1. Invasive Drone Encroachment Neutralization Techniques in DTS

The maintenance and security of drone transportation present a variety of surveillance issues due to the continued advancement of the drones’ underlying technology [1,20]. To conduct drone surveillance, it is necessary to communicate with, identify, authenticate, elicit, and disarm drones considered dangerous from a drone pool in a DTN. Effective drone surveillance entails the convergence or fusion of many technologies that work together for a specific purpose. Hence, the breakdown of the entire system implies a failure of the underlying technology in any of its components. As shown in Figure 1, a variety of methods (radio frequency, radar, thermal, acoustic, vision, and sniffing) are used to determine the location of a drone, the timing of its entry into a spatial area, and the appropriate divergent action (disable, disarm, or destroy) to take to keep the UAV within an authorized jurisdiction or destroy it [5,21,22]. Only the vision-based approach of these can provide an accurate visual description of the drone and the conveyed object with attendant weaknesses, which is essential for selecting the appropriate neutralizing response [23].

### 2.2. Collaborative Counter-Invasive Encroachment Approaches in DTS

Modern hybrid counter-invasive encroachment techniques combine sensor technology with hardware control systems to overcome the drawbacks of each detection method, expand the scope of detection, and enhance decision-making ability.

The use of multiple radio frequency (RF) scanners in detecting drones is prevalent these days due to its lower cost compared with radar detection technologies [3]. With multiple RF fusion detection technology, the control commands and other crucial data about multiple tracked drones and their operators in the network can be retrieved. However, newer drone designs typically outperform this sensor fusion surveillance strategy. A vision-acoustic sensor fusion strategy tends to improve detection precision by leveraging exact representation and long-range detection [24]. The effectiveness of the method depends on the auditory signals’ resistance to weather and other external factors as well as the precise visual description of the identified objects generated by electro-optical cameras. The inherent signal interference in a noisy environment is its drawback.

While many hybrid invasive drone encroachment and neutralization technologies integrate vision and radar, little attention is paid to integrating vision and light detection and ranging (LIDAR). Both RADAR and LIDAR detection aim at wide-area aerial object detection and identification. However, while RADAR uses radio waves to detect objects, LIDAR makes use of light waves. RADAR can detect objects at a distance of up to 30 km, but its capacity to do so is constrained by the size of its antenna. On the other hand, LIDAR offers 3D mapping with a high detection resolution accuracy of airborne objects and is the ideal alternative for a portable solution, which informs the deployment of LIDAR for 3D point cloud image generation. According to the authors in [25], LIDAR can now function at wavelengths above 1400 nm with a 500 m object detection range, 10% object reflectivity, and a 99% recognition confidence level. Consequently, recognition is a more difficult challenge than detection because it depends on the fine resolution and precision of real-time image processing, which is made possible by deep learning. Therefore, the drawbacks of the single application of each of these detection and recognition technologies can be addressed by integrating vision with the LIDAR approach, assisted by an AI learning model.

In reality, most drone surveillance architectures and configurations use network-based symbiotic multi-dimensional surveillance system implementations that rely on trustworthy AI models and data as well as operate on low-latency networks to make up for shortcomings in the current sensor fusion drone detection technology, as shown in Figure 2. Obtaining trustworthy data is currently difficult in this area. When collaborative learning is involved, the quality of decisions, the privacy of data, and the timeliness of responses become critical success factors in this mobile cyber–physical system. As a result, this paper’s goal is to enhance the existing vision-LIDAR fusion architecture to increase its detection speed, recognition accuracy, and range estimate. To do this, we developed a reliable drone-based dataset, proposed an effective underlying image-lidar-based detector, and formulated an assisted-learning strategy to determine the degree of invasion, evaluate the drone’s proximity to the targeted area, ascertain the impact of its threat(s), and choose the response strategy to use. This method of drone invasion defense will help the drone transportation system (DTS) become more acceptable and sustainable as a potential future “just-in-case” airborne vehicle type in smart cities.

## 3. Materials and Methods

### 3.1. ALIEN Design for Secured DTS

Managing complex and dynamic activities in real-time system designs demands both swiftness and accuracy. AI and system autonomy are inextricably linked. A scenario-based, adaptive-conscious, cognition-friendly AI model is at the heart of a data-centric, hard-based, real-time cyber–physical system. The ALIEN scheme therefore predicates and perpetuates situation-aware safe-channel neutralization, effective detection of variant UAVs, accurate visual identification of conveyed objects, and a timely interception in clusters, as conceptualized in the block diagram in Figure 1 and the system flowchart in Figure 2 respectively.

The objective function of the ALIEN scheme (as seen in Equation (Equation 1)) is an optimal solution to maximize DTS counter-invasive deployment through efficient resource utilization that is subject to time.

Mathematically, this is expressed as;
(1)$$I_{max}=t_{i}D_{i}+t_{i}{D_{r}^{d}}_{\lambda_{i}}+t_{i}D_{N_{i}};$$
where $I_{max}$ = objective function for the optimal solution; $D_{i}$, ${D_{r}^{d}}_{\lambda }$, and $D_{N}$ = decision variables representing drone detection (see Section 3.3), attached object recognition (see Section 3.5), and neutralization (see Section 3.7); subject to $t_{i}$, the time taken to perform each of these tasks for each *i*th drone in the DTS network. Performing these tasks with a centralized procedure and feedback will imply poor timing or a delayed, counter-productive response.

There are parallel activities for detection and identification before neutralization (the ultimate decision). Identification involves performing a threat analysis to determine the position, legality, and danger of the drone and the object it is carrying. The model and associated item of the drone are determined by the detection task using morphological features. The system takes in drone images and coordinates, performs pattern discovery using the underlying detector, produces output, feeds the output for further analysis, and chooses the best response strategy to use using heterogeneous sensors (electro-optical camera and topographic lidars), databases, networks, etc. The robustness of the system is measured by how effectively it can distinguish between different drone types and sizes with discrete attached objects when operating in sunny, cloudy, and evening climatic conditions.

### 3.2. Camera-Lidar Sensor Calibration

For airborne object detection and ranging operations, an Ouster OSI-128 lidar and a digital camera (Logitech C922) are used. As shown in Figure 1, the lidar uses an infrared source to sense an object’s movement, track its direction and speed, and determine its elevation from the ground network. To calibrate the camera and lidar, it is necessary to determine how the sensors are oriented and placed relative to one another. These calibration parameters are transformed into sensor measurement data, which is essential when updating the coordinate system. Geometrical and optical characteristics of the camera, such as its distortion coefficient, primary point, and focal length, are required to estimate the camera’s translation and rotation with respect to the lidar. Using the camera calibration approach, these features can be approximated. To estimate the collection of feature points, this work uses a checkerboard target [26]. In calculating the camera’s intrinsic matrix, each feature point is related to many angles. The perspective-n-point (PnP) technique then obtains the extrinsic camera and lidar parameters.

The PnP algorithm minimizes the reprojection error in pose estimation between the corresponding 2D points in the camera image and the corresponding 3D points in the lidar. The appropriate 3D–2D points are carefully chosen by using the reflective map of the lidar measurements from several checkerboard targets. As a result, more calibration accuracy than when employing a single plane is recorded.

### 3.3. ALIEN Detector and Ranging Estimation (D)

In all operational environments, an effective drone and range detector must distinguish drones from similar flying objects and offer a reliable range estimate of their distance from the detecting device. To do this, Figure 3 presents the proposed heterogeneous drone detection, range logic, and fusion procedure.

Object detection is carried out on the aerial images acquired by the electro-optical camera using the efficient drone detector network (ALIEN) model. To combine detection performance speed and accuracy in real-time, the ALIEN is an improved version of the DRONET [5] detector model that incorporates strip networks (SPP), focus, path aggregation networks (PANET), and other technologies, as illustrated in Figure 4.

The improvement in the detection model is the addition of extra bottleneck CSP layers at the backbone and neck layers, as captured in Table 1. Using a secure networked electro-optical camera, the “Backbone” receives the collected drone image (input), where feature extraction is carried out using a cross-stage partial network (CSP). After entering the “Neck” with PANET and the feature pyramid network (FPN) for feature fusion, the “Head” produces the real detection results, which include the position, score, size, and class. The bottleneck CSP at the backbone layer, combined with SPP and focus, reduces the complexity of large gradient information, truncates the gradient flow of the optimized network, and preserves feature extraction accuracy, as shown in Figure 5.

To do this, CSP separates the feature map of the base layers into two. While the first enters a dense block, the other gets integrated with the feature map and transferred to the next stage. The feed-forward propagation and weight update for this process are captured in Equations (Equation 2) and (Equation 3).
(2)$$Y\left\{\begin{array}{c} Y_{k}=W_{k}*\left[Y_{0^{\prime \prime }},Y_{1},...,Y_{k-1}\right]\\ Y_{t}=W_{t}*\left[Y_{0^{\prime \prime }},Y_{1},...,Y_{k}\right]\\ Y_{u}=W_{u}*\left[Y_{0^{{}^{\prime }}},Y_{t},\right] \end{array}\right.$$
(3)$$W\left\{\begin{array}{c} W_{k^{{}^{\prime }}}=f\left(W_{k},gi_{0^{\prime \prime }},gi_{2},gi_{3},...,gi_{k-1}\right)\\ W_{k^{{}^{\prime }}}=f\left(W_{k},gi_{0^{\prime \prime }},gi_{2},gi_{3},...,gi_{k}\right)\\ W_{u^{{}^{\prime }}}=f\left(W_{u},gi_{0^{{}^{\prime }}},gi_{t}\right) \end{array}\right.$$
with $Y_{k}$ being the input of the (k+1)th at the dense layer, gi being the network gradient information, and *W* being the weight. The backbone produces output with fewer channels, layers, and larger images.

Then, to further reduce the information path, enhance feature pyramid operations, and boost image localization accuracy, image instance segmentation is applied at the neck layer using PANET, which comprises four cardinal procedures. Table 1 summarizes the ALIEN convolutional structure, listing the specific convolutional parameters and values of its components.

First, a bottom-up path aggregation operation is performed to improve the localization capabilities of the feature extraction hierarchy by dispersing the low-level patterns. Then, feature levels with the same spatial size that corresponds to $P_{2}$ to $P_{5}$ are produced using bottom-up augmentation. Second, an new feature map with scales ranging from $N_{2}$ to $N_{5}$ is created. Each $N_{i}$ enters the convolution blocks in phases to further shrink the spatial size of the input. A fused feature map is then produced by fusing the reduced map and each $P_{i}$. This enters one more convolution block and generates a new $N_{i}+1$ for the subsequent sub-network, which leads to the third step, adaptive feature pooling. To achieve adaptive pooling, each proposal is assigned to a different feature level. Then, the fourth operation is fully-connected fusion. Fully connected fusion is carried out by utilizing region of interest (ROI) alignment on the feature grids. ROI alignment distinguishes the properties of the foreground and background masks of the input by training more sample aerial images using the parameters of the fully linked layers, which makes actual prediction possible at the head component. Particularly for small objects, PANET dramatically improves the image feature process.

At the head layer, actual prediction outcomes are performed based on the neck layer’s characterization (drone, attached object, etc.). The YOLO head of the detector model applies the CNN concept to detect objects by using a single network to divide the image into regions of interest (N*N grid), as seen in Figure 6.

Then, using the formula in Equation (Equation 4), the “head” predicts each bounding box region and probability before calculating the overlap known as the intersection of union (IOU).
(4)$$\begin{array}{c} \Rightarrow I_{ou}=\left[A_{\cap }*\frac{1}{A_{\cup }}\right]; \end{array}$$
where $A_{\cap }$ = area of intersection, and $A_{\cup }$ = area of union. The $I_{ou}$ value ranges between 0 and 1 with a threshold >0.5. To enhance the performance of the traditional YOLOv5 model’s aerial object recognition and prediction, the highlighted network configuration settings in Table 1 are the key modifications made.

Thereafter, the PnP technique [27] is employed to extract global and local characteristics from the 3D points to acquire the visual aerial object detection from the ALIEN detector model with its associated lidar measurements. It is not essential to perform the vertical binning required for other representations because PnP converts 3D points into pillar representation. The 3D point target from the PnP algorithm and the 2D drone target from the ALIEN model is then transformed into a single coordinate system using the sensor calibration parameters, and the point clouds are then projected onto the detected drone images to represent the range estimate, object detection, and identification ($D_{r}$) as shown in Figure 3.

### 3.4. Assisted Learning for Invasive Drone Interception $\left(D_{d}\right)$

To intercept invasive drone encroachment in an aerial transportation network, a cognitive networked-based real-time decision is taken by carrying out perceived danger or threat analysis to identify and intercept a perceived malicious drone or drones from among other drones and aerial objects in an aerial transportation network. Precision and promptness are essential elements.

**Theorem** **1.**
*Given that there are $n_{d}$ numbers of drones in a DTN, the estimated danger ($D_{r}^{d}$) to be carried out by a particular presumed malicious drone ($D_{r}$) in the network is a function of the union of the defined metrics and elements in the universal set of its operation as expressed in Equation (Equation 5):*

(5)
$$D_{r}^{d}=f\left\{D_{o}\cup D_{f}\cup D_{l}\right\},$$

*where $D_{r}^{d}$ = overall drone’s perceived danger,*

*D_l_ = legality determinant based on proximity and range measurement,*

*D_f_ = quantifier for physical feature of the drone, and*

*D_o_ = conveyed object characteristics.*


All transmitted messages in the DTN must be operated within a declared maximum load ($L_{max}$), have a defined priority ($k_{p}$), and be delivered within a set time ($t_{p}$) since drone interception needs real-time communication among the participating nodes. Thus, minimizing detection error ($|\left(t_{p+1}\right)-\left(t_{p-1}\right)|$), protocol latency ($t_{p+1}$), and transmission delay ($t_{p-1}$) is crucial in the network for effective interception. Based on the output of the model and related network media outputs, each of these parameters is evaluated. The procedures for intercepting a drone that is believed to be harmful in a DTN are outlined in Algorithm 1 with the drone-report as the expected output.
**Algorithm 1:** Steps for Invasive Drone Interception $D_{r}^{d}$.
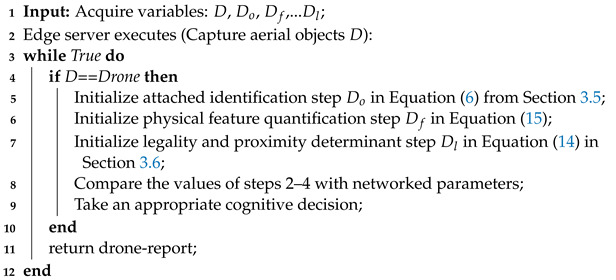


### 3.5. Harmful Object Recognition ($D_{o}$) and Feature Quantification ($D_{f}$)

An inappropriate sense of panic can be generated by the sight of a UAV with a strange object attached. As shown in the network diagram in Figure 7, simultaneous visual identification of a drone and the object it is conveying in a DTN is a challenging object detection task (an NP hard issue).

The detection devices (A, B, and C) in the DTN distinguish between different detected aerial objects and choose the best state of each object based on dynamic characteristics. In AL, each cluster for invasive drone encroachment (say, device A) analyzes the drones within its sensing range (a,1; a,1; a,0; a,0; a,0) based on its unique characteristics. Then, with the help of its underlying detection model, it carries out feature extraction and learns from the detected object pattern. Then, the hub shares and communicates the learned or acquired knowledge with other clusters (devices B and C) in the DTN. With subsequent detection and learning, the acquired knowledge from each node is constantly updated in the DTN by the participating nodes/clusters thereby fostering connected intelligence for timely decision making. Therefore, this precise collaborative identification ensures proper and faster eliciting of hostile drones attempting to encroach into the network and differentiates them from hobbyist or logistics drones, thereby gauging their perceived threat in any given setting. To analyze this multivariate attached object recognition and elicitation scenario problem, we use Theorem 2.

**Theorem** **2.**
*Given that a detected drone is elicited to be malicious ($D_{r}^{d}$), the dynamic estimate of its threat (${D_{r}^{d}}_{\lambda }$) in a given environment is a measure of its attached object characterization (${D_{r}^{d}}_{o}$), the technique for object detection (${D_{r}^{d}}_{f}$), path planning/routing (${D_{r}^{d}}_{path}$), and variability of its time (${D_{r}^{d}}_{time}$). This is expressed as Equation (Equation 6):*

(6)
$${D_{r}^{d}}_{\lambda }=f\left[{\left(\left({D_{r}^{d}}_{o}\cup {D_{r}^{d}}_{f}\right)\cup {D_{r}^{d}}_{path}\right)}^{{D_{r}^{d}}_{time}}\right],$$

*where ${D_{r}^{d}}_{o}$= $O_{t}$ = conveyed object threat induced by a drone represented as (a,1), (b,1), and (c,1);*

*${D_{r}^{d}}_{f}$= other physical feature quantifier = q;*

*${D_{r}^{d}}_{path}$ = the drone’s flight path = p; and*

*${D_{r}^{d}}_{time}$ = response time = t, represented in Figure 7 by (a, b), (a, c), and (b, c). The maximum value of ${D_{r}^{d}}_{time}$ is set at α, β, and 1. In this study, less attention is given to ${D_{r}^{d}}_{path}$ and ${D_{r}^{d}}_{time}$.*


The other physical feature quantifier’s (q) estimation is dependent on the underlying detection technology deployed. Therefore, a malicious drone’s ${D_{r}^{d}}_{f}$ value can be a combination of weight/kinetic energy ${D_{r}^{d}}_{\left(o,k\right)}$ = k, noise level ${D_{r}^{d}}_{\left(o,n\right)}$ = n, loaded object ${D_{r}^{d}}_{\left(o,o^{\prime }\right)}$ = o, and scanability ${D_{r}^{d}}_{\left(o,s\right)}$ = s values as expressed in Equation (Equation 7).
(7)$${D_{r}^{d}}_{f}=\left(k,n,o,s\right),$$

Generally, the mathematical formulae for deriving the loaded object (o), i.e., $D_{o,o^{\prime }}$ from an electro-optical sensor reading is given by Equation (Equation 8):(8)$$\begin{array}{c} {D_{r}}^{d_{\left(o,o{\;}^{\prime }\right)}}=o=\left[\left(\frac{level}{4}\right)*W_{l}\right], \end{array}$$
where level is the drone’s proximity to the field of view as measured by the lidar point cloud, and $W_{l}$ is the perceived weight of the object being transported. For instance, prompt action and caution are triggered to reroute the drone’s movement if the recognized object (${D_{r}^{d}}_{\left(o,o^{\prime }\right)}$ = o) is found to be potentially dangerous and inside the line of sight. In general, the dynamic threat estimate of a malicious invasive drone in a DTN is determined by substituting Equation (Equation 6) for the enlarged expression of $D_{f}$ from Equation (Equation 7).
(9)$${D_{r}}^{d_{\lambda }}=\left\{\begin{array}{c} \begin{array}{c} \left(\left((O_{t}\cup \left(k,n,o,s\right)\right)\cup {D_{r}^{d}}_{path}{)}^{{D_{r}^{d}}_{time}}\right)*N_{d}; \end{array} \end{array}\right.$$
where $N_{d}$ = number of swarming drones with conveyed objects within line of sight. However, keep in mind that the loaded object, ${D_{r}}^{d_{\left(o,o{\;}^{\prime }\right)}}$, is a subset of ${D_{r}^{d}}_{o}$, i.e., *o*⊆$O_{t}$. As a result of the law of association, Equation (Equation 9) is transformed into Equation (Equation 10):(10)$${D_{r}^{d}}_{\lambda }=\left\{\begin{array}{c} \begin{array}{c} \left({\left((k,n,o,s\right)\cup p)}^{t}\right)*N_{d}; \end{array} \end{array}\right.$$

Each of these factors in a hybrid anti-drone model is obtained from several sensor metrics of the underlying detecting technology (video, acoustic, thermal, thermal, radio frequency, etc.). Because this strategy is vision-based, only the values used in deriving $D_{\left(o,o{;}^{\prime }\right)}$ ($O_{t}$) can be determined from the electro-optical and lidar range sensor readings. Equation (Equation 10) is so changed to:(11)$$\begin{array}{c} {D_{r}^{d}}_{\lambda }=f\left[{\left(\left(O_{t}+\Theta \right)\cup p\right)}^{t}\right]*N_{drones}; \end{array}$$
where Θ represents other physical feature quantifiers associated with other detection technologies. In addition, *p* and *t* are not covered as they are outside the scope of this study. Thus, the total drone threat value is:(12)$$\begin{array}{cc} ∴\Sigma_{i}^{N_{d}}D_{T}= & max[\alpha \left({D_{r}}^{d_{\lambda }}\right)]; \end{array}$$
where max and α values represent the maximum allowable drone’s conveyed object threat value; $n_{d}$ is the number of detected drones with conveyed objects. Algorithm 2 highlights the steps for attached object identification and elicitation in a DTN.
**Algorithm 2:** Harmful Object Recognition and Elicitation (${D_{r}}^{d_{\lambda }}$).
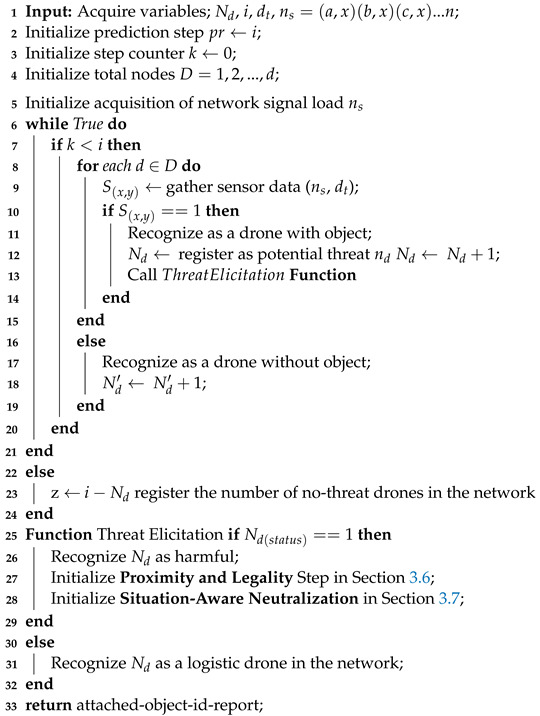


### 3.6. Assessing Drone Encroachment Legality ($D_{l}$)

Aside from determining the harmful status of DTS insight based on object characteristics (see Equation (Equation 6)), we verify the drone’s encroachment status. The assumption that several UAVs operating in a specific mapped area are given true flight paths is kept. Adherence to defined pathways, aside from authentication, helps establish whether an incoming drone’s flight is legal and permitted to enter a restricted area.

Mathematically, Equation (Equation 13) defines whether or not a drone is legal.
(13)$$D_{l}=f\left[A_{p}\cup A_{k}\right],$$
where $A_{p}$ = area of interest/the mapped area priority level, and $A_{f}$ = authorization permit/authentication key to operate within such a classified zone. Authenticating a drone’s authorization status in a DTN is an emerging research issue [28,29,30].

Mathematically, $A_{p}$ estimates the distance between the target drone and the predetermined restricted zone as expressed in Equation (Equation 14):(14)$$\Rightarrow A_{p}=L_{b}=\left[\frac{1}{1+exp(d_{a}-\frac{D_{max}}{2})}\right],$$
where $L_{b}$ = the detection range/legality boundary, $d_{a}$ is the distance between the area of interest and the detected drone, and $D_{max}$ = the system’s allowable detection range based on the sensor (it is usually a constant value). An environment that is categorized or a significant spatial domain is indicated by a high-priority area value. Active and suitable neutralizing decisions are necessary as the drone moves closer (based on measurements from the point cloud’s lidar range). The step-by-step procedure for determining a drone’s closeness, legality, and authorization is summarized in Algorithm 3.
**Algorithm 3:** Proximity and Legality Assessment ($D_{l_{k}}$).
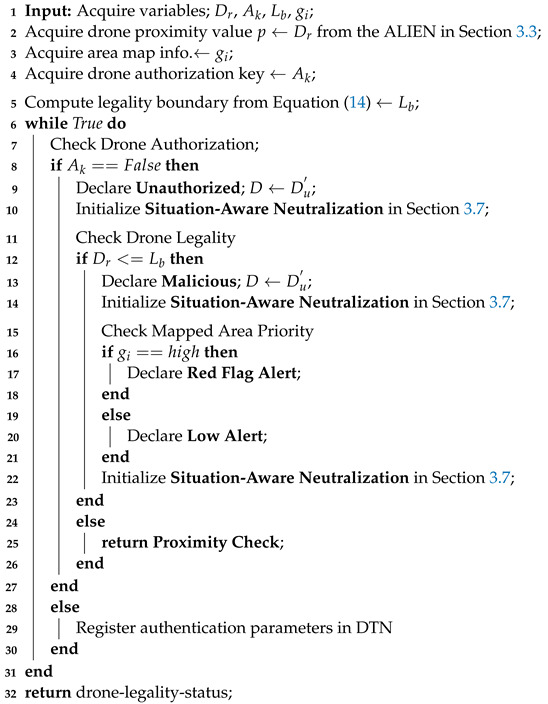


### 3.7. Adaptive Neutralization ($D_{N}$)

In a real-time control system routine, an adaptive and cognitive response feature is essential to avoid the deployment of flawed regimens or schedulers that use preset routines and set off false response alerts. Before instantiating and responding to a given scenario, an adaptive neutralization strategy should:Detect the peculiarity of the aerial object and its ranging measurement: drone ($D_{r}$) or not drone, ($D_{r}^{\prime }$) (see Section 3.3);Identify the conveyed object: attached objects ($D_{o}$) or no-attached-object, (${D_{o}}^{\prime }$) (see Section 3.5);Access the harmfulness of conveyed objects: harmful ($D_{T}$) or not harmful, (${D_{T}}^{\prime }$), through threat analysis (see Section 3.5);Determine the legality of the drone’s route: legal ($D_{l}$) or not legal, (${D_{l}}^{\prime }$) (see Section 3.6); andAscertain the authorization to operate: authorize ($A_{f}$), or not authorize, (${A_{f}}^{\prime }$) (see Section 3.6),

Before taking the appropriate neutralization approach: destroy (DefenseMode), disarm (SafeMode), or direct (Re−routeMode). To analyze the neutralization response scenario, we use Theorem 3.

**Theorem** **3.**
*Assume that a given drone model ∂D, conveying an identified object $\partial D_{\left(o,o^{{}^{\prime }}\right)}$, with a classified status $\partial D_{\left(T,T^{{}^{\prime }}\right)}$, flies into an environment based on legality route $\partial D_{\left(l,l^{{}^{\prime }}\right)}$, and has authorization key to operate $\partial A_{\left(f,f^{{}^{\prime }}\right)}$, the objective function of the neutralization response provided that flight path $D_{path}$ is known and response time of the system, $D_{time}$ is swift and is given by Equation (Equation 11):*

(15)
$$\begin{array}{c} D_{N_{max}}=\left[\left(\partial D*\partial D_{\left(o,o^{{}^{\prime }}\right)}*\partial D_{\left(T,T^{{}^{\prime }}\right)}*\partial D_{\left(l,l^{{}^{\prime }}\right)}*\partial A_{\left(f,f^{{}^{\prime }}\right)}\right)*N_{drones}\right];\\ s.t.D_{path},D_{time} \end{array}$$

*where $D_{N_{max}}$ represents the optimal solution for maximum neutralization action, representing the three (3) possible responses (destroy, disarm, or direct/re-route) depending on the dynamic value or state of $D_{N}$, and * is the weight multiplier effect of each counter-invasive activity subject to the time required to perform each activity and the path flight of the drone.*


The ALIEN technique ensures proactive and automatic drone involvement in the network at changing dynamic intervals based on extracted and acquired features, behavioral traits, and other networked system information instead of rule-of-thumb heuristics. Algorithm 4 summarizes this proactive and situation-aware neutralization reaction.

### 3.8. Dataset Collection, Characterization, and Preprocessing

To assess UAV encroachment, two (2) datasets are used for simulation purposes; one for invasive UAV detection and the other for attached object identification. We created the visioDECT dataset for UAV detection, available on the IEEE Dataport [31]. VisioDECT comprises 20,924 drone samples taken from six (6) UAV models. Each of these UAV types represents a superclass. Each of these three superclasses contains three (3) scenarios (cloudy, evening, and sunny) of flown drones that represent the subclass. Then, each subclass represents the individual drone samples of different sizes flown at various locations, with different distances and altitudes and at different times of the day. The attached object recognition dataset comprises 3600 samples from nine (9) attached objects mounted to drones to represent the classes. Table 2 summarizes the dataset description and distribution.
**Algorithm 4:** Pseudocode for Situation-Aware Neutralization ($N_{\left(d,s,r\right)}$).
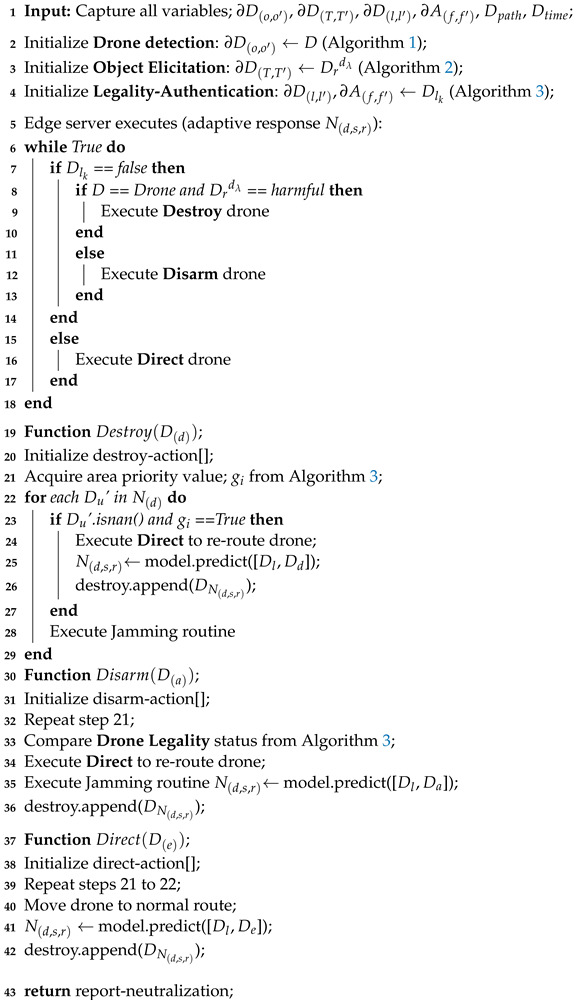


These drone types were chosen after considering research by [32] on their popularity in pertinent industries. Additionally, the need to consider both customized drones used in industry and by enthusiasts justifies the inclusion of these drones in the sample space. A detailed explanation of the visioDECT dataset generation and collected data can be accessed via [31]. The datasets were manually created by flying each UAV model under different weather conditions, at different times, and in different locations. As demonstrated in Figure 8, videos of the flown drones with attached objects (at heights ranging from 30 to 100 m) were captured using digital cameras and lidars.

Then, using the “Free Video to JPG” software, the captured video frames were converted into a series of sample frames. To ensure the accuracy of the data, the data frames were sorted to eliminate samples of frames with no drones in the background. Then, each sample frame was labeled by creating bounding boxes around the target objects. To generate ground truth values, bounding boxes were drawn around the target objects, as shown in Figure 6. This arduous AI labeling process was accomplished with the help of the “Make Sense” application. Only 7200 samples (34.4%) of the visioDECT were used for model simulation, as shown in Table 2 describing the sample size distribution of the dataset for the experimental setup.

### 3.9. Experimental Setup

For simulation, the datasets were split into three portions for model training, testing, and validation: 70%, 20%, and 10% to prevent model overfitting. All models were created using the same network depth-multiple and width-multiple values of 0.33 and 0.50, together with extra hyperparameters as given in Table 3.

Model ablation studies were carried out by varying the hyperparameters to validate the detection performance of the proposed model. To evaluate the capability of the proposed model for knowledge discovery and learning, transfer learning was implemented. The pre-trained weights were used to determine the point cloud for range measurement using PointPillars. Before model training, different image augmentations were applied to the images to lessen object misrepresentation. After that, the best and last weights were used to evaluate the inference of the model. The experimental simulation platform was carried out in a Python environment with PyTorch 1.10 framework on a computer running Windows 10 with the following specifications: Intel(R) Core(TM) i5-8500 CPU @ 3.00GHz, 6Core(s), NVIDIA GeForce GT 1030, GPU CUDA:0 (Tesla K80, 11441.1875MB), and 36GB RAM.

## 4. Experimental Results and Discussion

The experimental findings are presented in this section to assess how well the suggested ALIEN works to identify and elicit harmful drones and attached objects in a DTN. Additionally, a thorough performance evaluation comparison of the ALIEN model and SOTA approaches were performed using performance metrics, such as mean average precision (mAP), specificity, sensitivity, F1-score, G-mean, throughput (number of floating point operations per second (FLOPS)), and latency/response timeliness (frames per second). A further evaluation of the reliability and efficiency of the proposed model was performed to demonstrate that the scheme meets the criteria for a timely and efficient counter-invasive response.

### 4.1. Invasive Drone Detection and Elicitation by the ALIEN Model

Based on the visioDECT dataset [31], the findings in Table 4 illustrate the effectiveness of the ALIEN model for efficient drone detection and classification across scenarios.

Mean average precision (mAP) as a metric assesses the capacity of a model for accurate positive prediction and elicitation, expressed in Equation (Equation 16) as;
(16)$$\begin{array}{c} mAP=\left[\frac{\sum_{i=i}^{k}Ave.P_{r}\left(i\right)}{K}\right], \end{array}$$
with *K* = number of samples in the dataset, and $Ave.P_{r}\left(i\right)$ = average precision of each *i* sample. From Table 4, ALIEN achieved on average a 99.5% *mAP* value in detecting all UAV models for the cloudy and sunny scenarios, implying a high-level positive prediction capability. In addition, a closer look indicates that ALIEN performed relatively poorly in the evening due to obscured vision, with the lowest mAP of 12% for “Mavic-Air“ drones. Notwithstanding, a 99.5% mAP achieved in detecting miniature “Anafi-Ext” drones in the evening gives credence to the detection precision of ALIEN.

Sensitivity ($R_{c}^{+}$) measures the ability of a model to make accurate positive predictions. That is, the likelihood of a positive test reflects how well the model picks up true positives, written as seen in Equation (Equation 17):(17)$$\begin{array}{c} Sensitivity\left(R_{c}^{+}\right)=\left[t_{p}\times \frac{1}{t_{p}+f_{n}}\right], \end{array}$$
where $t_{p}$ = true positive predictions, and $f_{n}$= false negative predictions by the model, respectively. A high sensitivity value denotes a good model performance. According to Table 4, the sensitivity value of the proposed model ranges from 75% to 100%, implying that the ALIEN model accurately detects various drones in different climatic conditions and at varying altitudes.

From Figure 9, a 100% sensitivity value achieved by the ALIEN model in detecting a distant and miniature drone, Anafi-extended in the evening scenario is remarkable.

Furthermore, to assess the true negative prediction capacity of the ALIEN model for accurate invasive drone elicitation from other aerial objects, specificity ($R_{c}^{-}$) is expressed as by Equation (Equation 18);
(18)$$\begin{array}{c} Specificity\left(R_{c}^{-}\right)=\left[t_{n}\times \frac{1}{t_{n}+f_{p}}\right], \end{array}$$
with $t_{n}$ being true negative errors in prediction (i.e., returning a right signal that the detected aerial object is not a drone), and $f_{p}$ representing false positive errors in prediction (i.e., returning a wrong signal that an aerial object is a drone when it is not) by ALIEN model. A low $R_{c}^{-}$ value denotes good model performance. The low average $R_{c}^{-}$ value of 42.5% achieved by the ALIEN model (see Table 4) indicates the capacity of the model to handle the difficult task of simultaneous detection and elicitation of multiple aerial objects in a DTN.

The moving average results for the evening and cloudy climatic conditions in Figure 10 reveal that the ALIEN model can sufficiently decide that the detected aerial object is not a drone ($\partial D^{{}^{\prime }}$) even in an obscure scene, which is necessary to prevent unjustified interruption of the airspace.

Finally, to further examine the prediction characteristics of the ALIEN model, particularly when there is an uneven distribution in the variability of the sample size of the dataset, we evaluate a trade-off between sensitivity ($R_{c}^{+}$) and specificity ($R_{c}^{-}$), otherwise called the geometric mean (G-mean). The mathematical definition of G-mean is:(19)$$G-mean={\left({R_{c}}^{+}\times {R_{c}}^{-}\right)}^{\frac{1}{2}},$$

The results in Table 4 affirm that the low G-mean value (between 26.01–70.67%) recorded by the ALIEN model indicates good prediction performance when there is an uneven sample distribution. At a glance, the result from Figure 11 affirms the prediction ability of the proposed model with an uneven sample distribution.

### 4.2. ALIEN Learning Capacity through Ablation Study

Finding a balance between the reliance of the model on batch gradient descent and stochastic gradient descent can determine the objectivity of the model in terms of error committal and accuracy for each full pass of the training algorithm (epoch) throughout the entire training set. To carry out the ablation study on the proposed model, the training set was divided into various mini-batch sizes and used to calculate the error and update its coefficients, as shown in Table 5. A mini-batch size ($s_{n}$) refers to the portion of the training dataset ($S_{N}$) that the CPU processes simultaneously. The training time increases with $s_{n}$ size.

The result of the ablation study shows a gradual decline in the error minimization by the ALIEN model, an indication of a good learning ability across all batch sizes (batch 8 = 0.037, batch 16 = 0.039, etc.), with little prediction error as the batch size grew at a specified epoch of 100 and learning rate of 0.005 as captured in the line graph of Figure 12.

The results from the performance evaluation analysis (sensitivity, specificity, irrational behavior (F1), etc.) in response to detecting drones and attached objects under dynamic environmental scenarios confirm the suitability of the ALIEN model as an efficient underlying model for multi-scale invasive drone and aerial object detection and status elicitation necessary for intercepting illegal drone operations in a DTN at different altitudes and distances, as shown in the detected samples in Figure 13 with their range estimates.

### 4.3. Attached Object Recognition by the ALIEN Model

In a DTN that is used for threat analysis and situation-aware neutralizing decisions, accurate and exact transmitted object identification and elicitation are essential to distinguish a hobby drone or logistics drone from a presumed malicious drone (see Section 3.5).

The values from Table 6 show that the ALIEN model can, to some extent, detect and differentiate various conveyed objects by each UAV type (see samples of recognized attached objects Figure 14) with mAP values ranging from 99.7% to 31.7%, maximum sensitivity value of 89.5%, and specificity value of 51.1%.

In addition, with a 99.7% accurate visual recognition of conveyed objects, $\partial D_{\left(o,o^{{}^{\prime }}\right)}$ (as shown in Figure 15), the harmful status or otherwise, $\partial D_{\left(T,T^{{}^{\prime }}\right)}$, of the targeted drone can easily be ascertained through proper threat analysis as detailed in Section 3.5. This result validates the capacity of the ALIEN model for simultaneous drone detection and attached object recognition.

### 4.4. Performance Evaluation

Rationality and timeliness in decision making are crucial elements in any time-sensitive and precision-driven real-time system (such as an anti-drone system). This section compares and evaluates the performance of the ALIEN model with different YOLO models and other SOTA DL models.

#### 4.4.1. ALIEN and YOLO variants

The results in Table 7 summarize the performance of the ALIEN model and variants of YOLO, highlighting the computational complexity analysis of each model.

The *F1*-score (F1) defined by Equation (Equation 20) measures the change in precision and sensitivity values of a model and quantifies how rationally the model behaves while performing a task.
(20)$$\Rightarrow F1=2\times \left[P_{r}*R_{c}^{+}*\frac{1}{P_{r}+R_{c}^{+}}\right],$$
where $P_{r}$ represents the precision of the model represented as mAP; and $R_{c}^{+}$ is the recall/sensitivity of the model. In ML, F1−score is preferred as a better performance evaluation metric than accuracy because an ML model needs to be rational while executing a task to minimize the false prediction or detection rate. The ALIEN model outperformed other YOLO models with F1-scores (F1) of 99.8% for drone detection and 80.1% for conveyed object recognition. This validates that the ALIEN model is more effective than other models for effective counter-invasive encroachment neutralization in a DTN.

The results in the line graph in Figure 16 show the mAP values of each model across the epochs, confirming the effectiveness of the ALIEN model for counter-invasive drone encroachment detection.

Furthermore, the latency of a model checks how quickly it responds to events (i.e., prediction time). By using the asynchronous execution method to measure delay, the results confirm that the ALIEN model outperformed other YOLO models with a latency value of 0.021s for both drone detection and conveyed object recognition.

#### 4.4.2. ALIEN and SOTA Model Performance

For exhaustive performance evaluation, the results in Table 8 highlight the computational complexity analysis of the ALIEN model and other SOTA models.

Firstly, throughput defines how frequently each model receives service requests or the maximum number of input instances a neural network can handle in a specific time. This is expressed mathematically in Equation (Equation 21);
(21)$$\Rightarrow \;Throughput\left(N_{t}\right)=\left[N_{b}*b_{n}*\left(\frac{1}{T_{t}}\right)\right],$$
with $N_{b}$ being the number of batches, $b_{n}$ is the batch size, and $T_{t}$ represents total time in seconds. According to Table 8, the ALIEN model achieved the best throughput of 16.1 GFLOPS which is closely followed by fpn+5v5 with 16.20 GFLOPS, YOLOv5 with 16.50 GFLOPS, and then MobileNet with 18.40 GFLOPS. However, overall, the ALIEN model has a better prediction performance than other SOTA models using other evaluation metrics, thereby making the proposed model a preferable choice model for counter-invasive encroachment.

Secondly, the dependability or reliability of a model is the measure of its error minimization, otherwise called loss. When compared to other SOTA models, the ALIEN model achieved the lowest loss (0.0370) in Table 8. Though the other YOLO variants exhibited similar low error minimization values (such as 0.0426 for YOLOv3, 0.0421 for fpn+v5, etc.), the ALIEN model still had the least loss value, signifying its reliability in prediction performance.

Thirdly, the efficiency of a real-time system is measured as the ratio or point at which its precision (mAP) coincides with its sensitivity ($R_{c}^{+}$) in carrying out a particular task.

Thirdly, the efficiency of a real-time system is measured as the ratio or point at which its precision (mAP) coincides with its sensitivity $R_{c}^{+}$) in carrying out a particular task. Equation (Equation 22) defines efficiency as;
(22)$$\Rightarrow Efficiency\left(\xi \right)=\left[\frac{P_{r}}{R_{c}^{+}}\right],$$

From the result in Table 8, v3 has $\left[\frac{97.20}{89.50}\right]$ξ value, v5s has $\left[\frac{96.50}{100.0}\right]$ξ value, p2 + v5 has $\left[\frac{98.30}{100.0}\right]$ξ value, fpn+v5 has $\left[\frac{100.0}{70.80}\right]$ξ value, etc. However, the ALIEN model achieved a $\left[\frac{99.50}{100.0}\right]$ξ value validating the model with the highest efficiency in handling multi-scale counter-invasive drone with aerial objects encroachment as well as conveyed object recognition prediction under dynamic scenarios. This prediction efficiency is further verified by the confusion matrix in Figure 17 and the detected samples in Figure 13 and Figure 14, with a minimal degree of misclassification.

### 4.5. Adaptive Encroachment Neutralization Analysis

According to the analysis of the experimental data presented above, the proposed model is capable of achieving high drone detection (*D*) values of 99.8% and effective conveyed object recognition ($D_{r_{\lambda }^{d}}$) value of 80.1%, which are required for perceived threat analysis, at a shorter response time ($D_{time}$) of 0.021s, which is required for prompt adaptive neutralization ($D_{N_{\left(max\right)}}$), and less memory consumption of 16.1%. The deployment of the ALIEN model as an underlying DL counter-invasion detection model in a mobile cyber–physical system operating with the hard-real-time control system principle can effectively carry out proper perceived threat analysis and drone authentication and trigger the appropriate neutralization strategy to stop or nullify a suspected malicious drone encroachment in a DTN before it disrupts the entire airspace transportation system, creates social apprehension, and creates public disapproval in an attempt to malign legal drone usage for critical emergency purposes and priority-based logistics.

The condition of a drone in the DTN can be determined at any time by combining these learnable DL model-based parameters (*D*, $D_{r_{\lambda }^{d}}$, $D_{time}$, and $D_{N_{\left(max\right)}}$) with other sensor-based and systems/network-derived parameters, such as legality authentication ($D_{l_{k}}$), flight path ($D_{p}ath$), map area priority ($g_{i}$), and the appropriate authentication security infrastructure. This guarantees that the evolving anti-drone control system can adeptly instantiate, identify, and elicit perceived malicious invasive drone encroachment from drifting hobby and logistics drones before launching an automated counter-response that could endanger the development of drone technology as a viable priority-based freight carrier. The sustainability of DTS is therefore guaranteed by this approach to invasive drone encroachment detection, identification, and neutralization because it permits only authorized drone-based logistics, airborne transports, and hobby drones to operate along the approved routes in the DTN while circumspectly observing premeditated malicious drone activities aimed at disrupting the network.

## 5. Conclusions

This work proposes a robust approach to ascertaining safety and security in a drone transportation network by circumspectly determining the malicious status of a drone in the network using a multi-modal deep-learning approach. The approach collaboratively detects and elicits the harmful status or otherwise of a drone in flight in a drone transportation network by detecting the drone in the network, identifying the conveyed objects, assessing the legality boundary of the drone, and determining its authenticity and authorization to operate before deciding the appropriate counter-invasive encroachment response to initiate based on given dynamic metrics and feedback parameters. Performance evaluation and comparison with nine other SOTA models were performed. The experimental results validate the adequacy and inventiveness of the proposed approach in ensuring security and sanity in a drone-based intelligent transport system for the viability and sustainability of DTS through objectivity and circumspection in decision making before interfacing with perceived targets.

Due to their miniature size and concealment of conveyed objects, low detection precision was observed in the prediction results. Future work will tackle this and other emerging problems by developing a drone-based swarm optimization algorithm to enhance the learning of the detection network model and improve performance. In addition, consideration will be given to drone authentication and authorization in a DTN using semi-blockchain technology and a functional non-fungible token to interface with the evolving drone technology landscape and its diverse usage as a viable intelligent vehicle for a just-in-case supply chain.

## Figures and Tables

**Figure 1 sensors-23-01233-f001:**
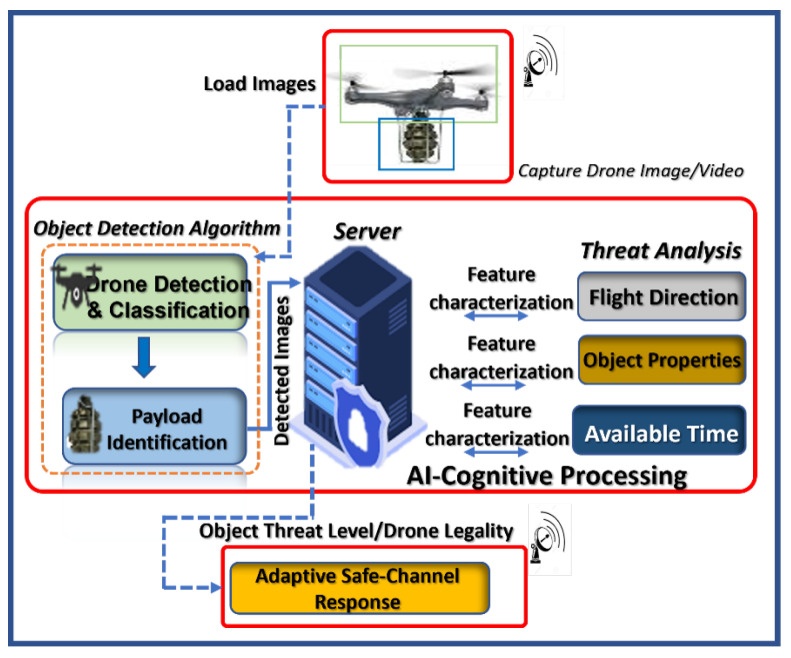
Architecture of collaborative cognitive invasive encroachment scheme highlighting the detection and classification, identification and unique recognition, threat analysis, and situation-aware neutralization interactions.

**Figure 2 sensors-23-01233-f002:**
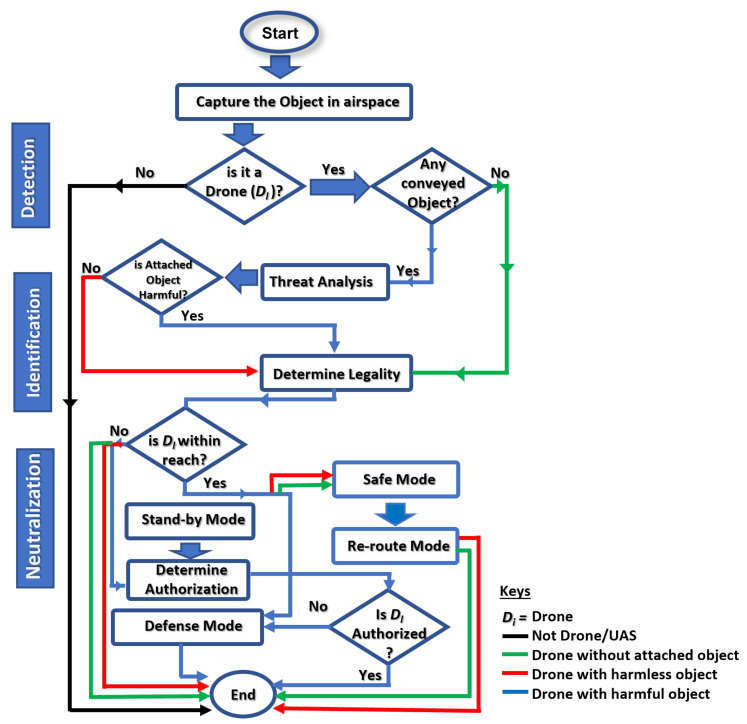
System flow of ALIEN procedure to cooperative counter-invasion in DTN.

**Figure 3 sensors-23-01233-f003:**
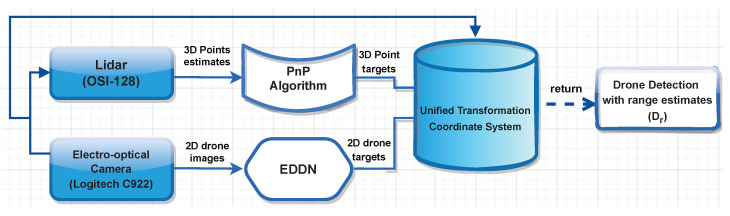
Logic flow of the efficient drone and ranging detector (EDRD) indicating how the drone images from the electro-optical camera and the ranging estimates from the lidar converge for efficient detection estimation and prediction.

**Figure 4 sensors-23-01233-f004:**
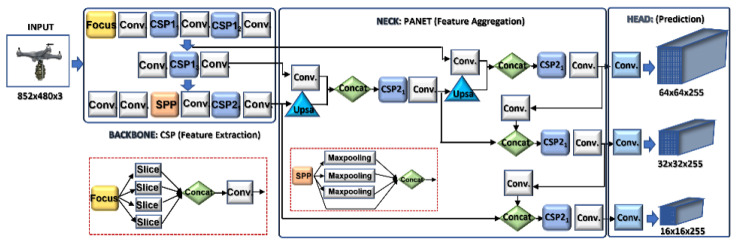
ALIEN model with its the underlying structures. The backbone layer comprises cross partial network (CSP) and focus for feature extraction. The neck layer consist of path aggregation network for feature aggregation, while the head layer has the YOLOv5 for actual prediction.

**Figure 5 sensors-23-01233-f005:**
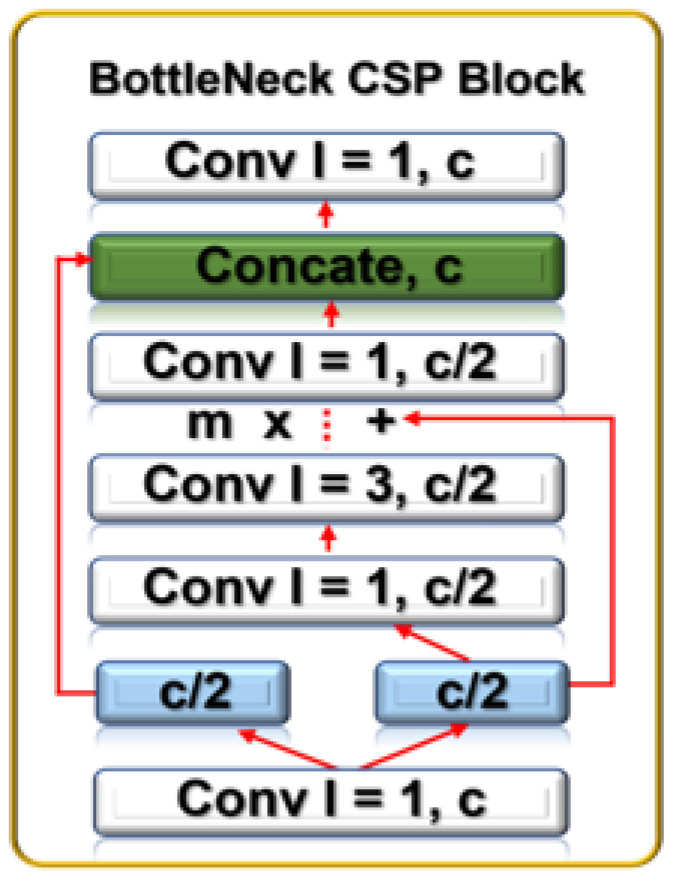
BottleNeck CSP block showing its constituent convolution blocks.

**Figure 6 sensors-23-01233-f006:**
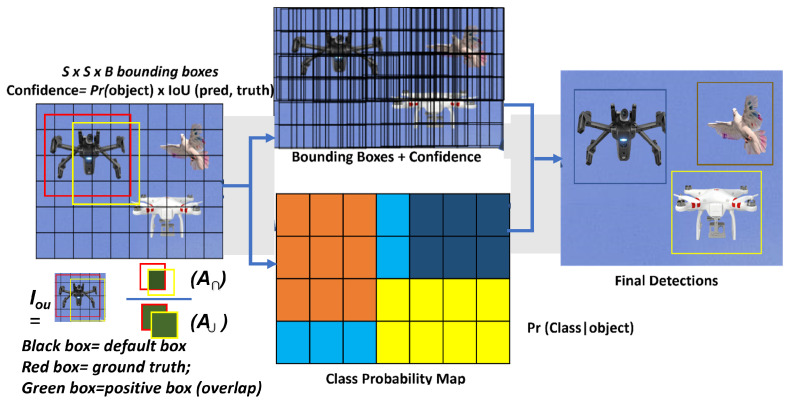
Drone feature extraction process highlighting the principles for determining outcome.

**Figure 7 sensors-23-01233-f007:**
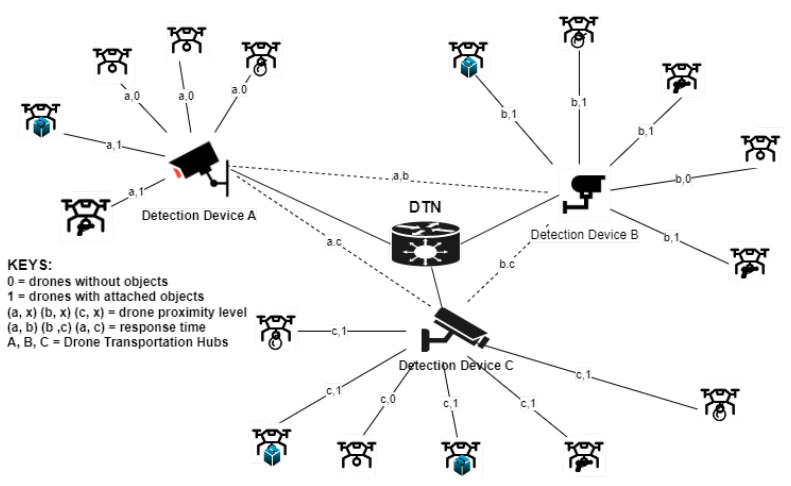
Assisted learning scheme for invasive encroachment neutralization in a DTN highlighting the various hubs of surveillance devices.

**Figure 8 sensors-23-01233-f008:**
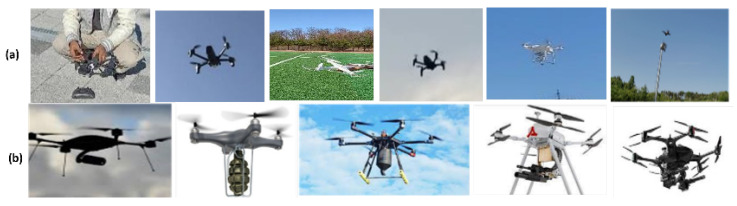
(**a**) Images of dataset capturing and flown drones at different altitudes and climate; (**b**) images of drones with attached objects.

**Figure 9 sensors-23-01233-f009:**
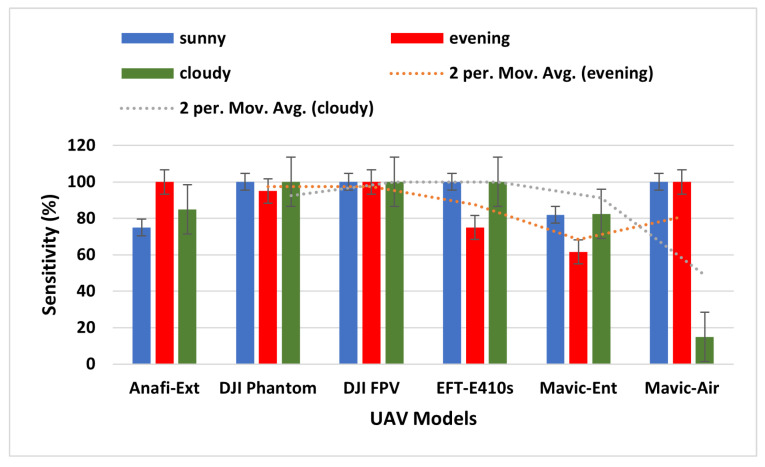
Sensitivity graph of ALIEN performance for detecting UAV models across all scenarios.

**Figure 10 sensors-23-01233-f010:**
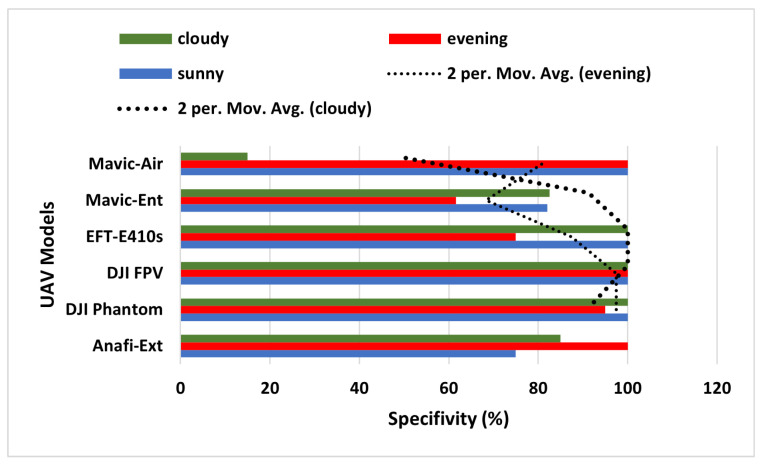
Specificity graph of ALIEN performance for detecting UAV models across all scenarios.

**Figure 11 sensors-23-01233-f011:**
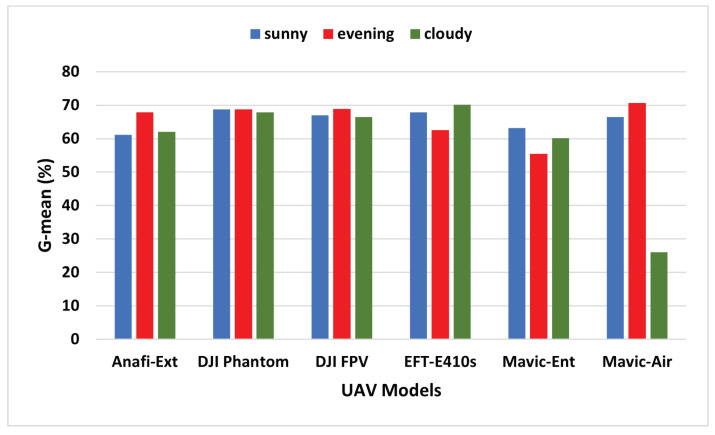
G-mean graph of ALIEN performance showing the trade-off between $R_{c}^{+}$ and $R_{c}^{-}$ in detecting different UAVs across all scenarios.

**Figure 12 sensors-23-01233-f012:**
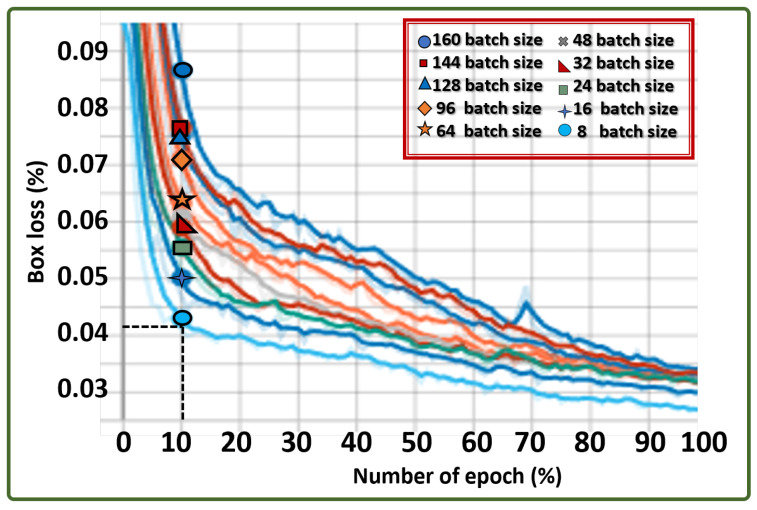
Box loss graph across different batch sizes indicating ALIEN’s learning capability in predicting results on different operation quanta.

**Figure 13 sensors-23-01233-f013:**
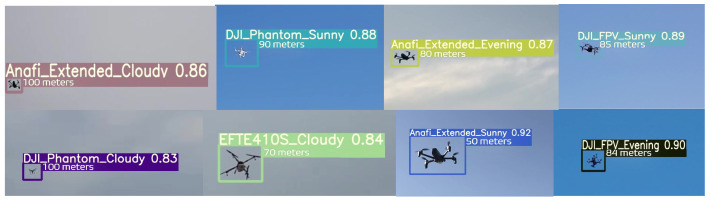
Drone detection samples by the ALIEN model across drone models at different heights and scenarios.

**Figure 14 sensors-23-01233-f014:**
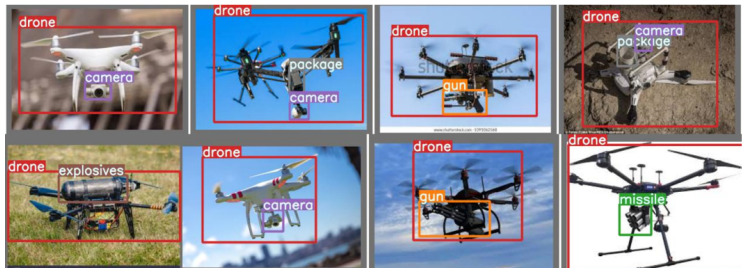
Samples of recognition of conveyed objects by ALIEN.

**Figure 15 sensors-23-01233-f015:**
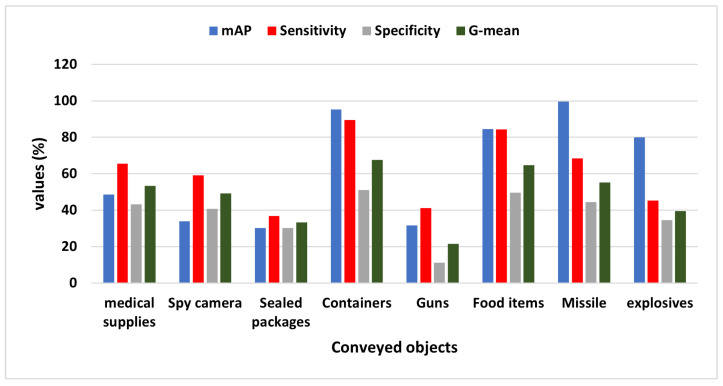
Graph showing the mAP, sensitivity, specificity, and G-mean values of ALIEN in identifying different conveyed objects by the drones.

**Figure 16 sensors-23-01233-f016:**
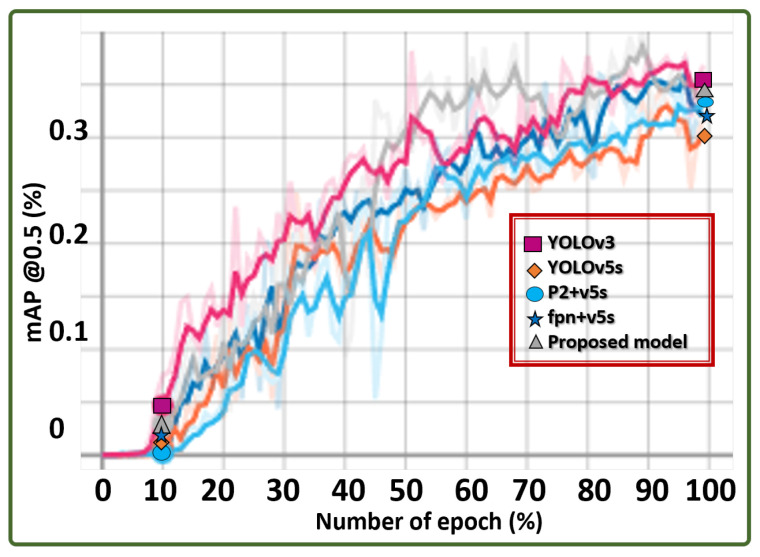
Hyperparameter graph showing the average precision of different models in identifying conveyed objects by UAVs.

**Figure 17 sensors-23-01233-f017:**
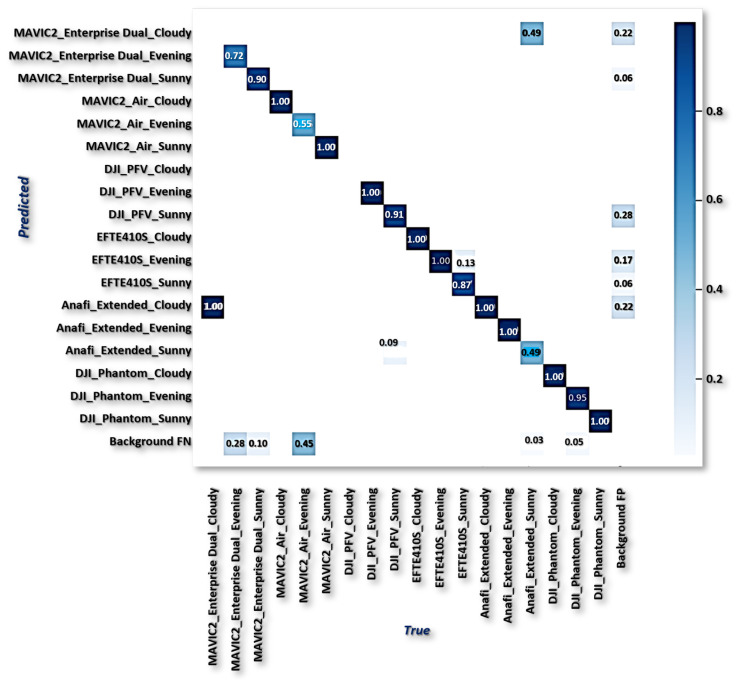
ALIEN confusion matrix.

**Table 1 sensors-23-01233-t001:** ALIEN convolutional structure.

ALIEN Model Configuration Specification		
**Layer**	**Output Shape**	**Descriptions**
depth_multiple	0.33	model depth multiple
width_multiple	0.50	layer channel multiple
	[−1, 1, Focus, [64, 3]]	
	[−1, 1, Conv, [128, 3, 2]]	
	[−1, 3, BottleneckCSP, [128]]	
	[−1, 1, Conv, [256, 3, 2]]	
	[−1, 9, BottleneckCSP, [256]]	
	[−1, 1, Conv, [512, 3, 2]]	
	[−1, 9, BottleneckCSP, [512]]	
	[−1, 1, Conv, [1024, 3, 2]]	
	[−1, 1, SPP, [1024, [5, 9, 13]]]	
**Backbone**	[−1, 3, BottleneckCSP, [1024, False]]	CSP performs the feature extraction on the acquired aerial images
	[−1, 1, Conv, [512, 1, 1]]	
	[−1, 1, nn.Upsample, [None, 2, `nearest’]]	
	[[−1, 6], 1, Concat, [1]]	
	[−1, 3, BottleneckCSP, [512, False]]	
	[−1, 1, Conv, [256, 1, 1]]	
	[−1, 1, nn.Upsample, [None, 2, `nearest’]]	
	[[−1, 4], 1, Concat, [1]]	
	[−1, 3, BottleneckCSP, [256, False]]	
	[−1, 1, Conv, [256, 3, 2]]	
	[[−1, 14], 1, Concat, [1]]	
	[−1, 3, BottleneckCSP, [512, False]]	
	[−1, 1, Conv, [512, 3, 2]]	
	[[−1, 10], 1, Concat, [1]]	
	[−1, 3, BottleneckCSP, [1024, False]]	
**Neck and** **Head**	[[17, 20, 23], 1, Detect, [nc, anchors]]	Image segmentation, Feature Aggregation, and eventual prediction are carried out

**Table 2 sensors-23-01233-t002:** Dataset Characterization.

Dataset Description				
**UAV** **Model**	**Scenario**	**Sample** **Size**	**Conveyed** **Objects**	**Sample** **Size**
Anafi-Ext	Sunny	200	Gun	400
	Evening	200	Medical Supplies	500
	Cloudy	200	Spy camera	400
DJI-Phantom	Sunny	200	Sealed Package	300
	Evening	200	Containers	350
	Cloudy	200	Food Items	500
DJI-FPV	Sunny	200	Explosives	240
	Evening	200	Missile	400
	Cloudy	200	Total Sample $\left(S_{N}\right)$	3600
EFTE410S	Sunny	200		
	Evening	200		
	Cloudy	200		
Mavic-Ent	Sunny	200		
	Evening	200		
	Cloudy	200		
Mavic-Air	Sunny	200		
	Evening	200		
	Cloudy	200		
Total Sample $\left(S_{N}\right)$		7200		

**Table 3 sensors-23-01233-t003:** Model hyperparameters.

Model Parameters		
No.	Parameters	Values
1	Batch size	8, 16, 24, 32, 48, 64, 128, 160
2	Box loss	0.05
3	Epoch	100
4	Input size	416 × 416 × 3
5	Learning rate	0.01 (0.005)
6	Weight-decay	0.0005
7	Warmup-epochs	3.0
8	Warm-momentum	0.8

**Table 4 sensors-23-01233-t004:** UAV model detection and classification performance.

Proposed Model Detection and Classification Performance					
**UAV** **Models**	**Scenario**	**mAP** **%**	**Sensitivity** **(**$R_{c}^{+}$**)** **%**	**Specificity** **(**$R_{c}^{-}$**)** **%**	**G-Mean** **%**
Anafi-Ext	sunny	99.50	75.00	49.86	61.15
	evening	99.50	100.00	46.10	67.89
	cloudy	99.50	85.00	45.20	61.98
DJI-Phantom	sunny	99.50	100.00	47.21	68.70
	evening	94.50	95.00	49.71	68.72
	cloudy	99.50	100.00	46.13	67.91
DJI-FPV	sunny	99.50	100.00	44.85	66.97
	evening	24.90	100.00	47.50	68.92
	cloudy	99.50	100.00	44.15	66.45
EFTE410S	sunny	99.50	100.00	46.12	67.91
	evening	44.60	75.00	52.25	62.59
	cloudy	90.80	100.00	49.13	70.09
Mavic-Ent	sunny	80.00	82.30	48.50	63.17
	evening	66.70	61.60	49.68	55.48
	cloudy	81.00	82.50	43.75	60.08
Mavic-Air	sunny	99.50	100.00	44.12	66.42
	evening	12.00	100.00	49.95	70.67
	cloudy	99.50	15.00	45.12	26.01

**Table 5 sensors-23-01233-t005:** Model ablation based on hyperparameter tuning.

ALIEN Model Ablation Performance.				
**Batch Size**	**Epoch**	**mAP** **%**	**Sensitivity** **(**$R_{c}^{+}$**)** **%**	**Box_Loss**
8	100	99.5	100	0.037
16	100	99.5	100	0.039
24	100	99.5	100	0.044
32	100	99.6	100	0.046
48	100	99.6	100	0.050
64	100	99.8	100	0.052
96	100	99.6	100	0.053
128	100	99.5	100	0.059
144	100	99.5	100	0.062
160	100	99.5	100	0.063

**Table 6 sensors-23-01233-t006:** Conveyed objects status recognition.

Recognition of Conveyed Objects by the ALIEN Model				
**Conveyed** **Objects**	**mAP** **%**	**Sensitivity** **(**$R_{c}^{+}$**)** **%**	**Specificity** **(**$R_{c}^{-}$**)** **%**	**G-Mean** **%**
Medical Supplies	48.5	65.5	43.3	53.3
Spy Camera	33.9	59.1	40.8	49.1
Sealed Packages	30.2	36.8	30.1	33.3
Containers	95.2	89.5	51.1	67.6
Guns	31.7	41.2	11.2	21.5
Food Items	84.5	84.3	49.6	64.6
Missile	99.7	68.5	44.5	55.2
Explosives	80.0	45.2	34.5	39.5

**Table 7 sensors-23-01233-t007:** Models performance evaluation I.

		UAV Detection Performance				Conveyed Object Recognition Performance			
**Models**	**Back**	**F1**	**Time**	**Train.**	**Space**	**F1**	**Time**	**Train.**	**Space**
	**Bone**	**(%)**	**(fps)**	**Time**	**GFLOPs**	**(%)**	**(fps)**	**Time**	**GFLOPs**
v3	Darknet	95.5	0.016 s	10 m 57 s	23.4	68.9	0.015 s	1 m 8 s	23.3
v5s	Tiny-s	98.1	0.021 s	5 m 27 s	16.5	61.5	0.022 s	2 m 54 s	16.4
p2 + v5	sPPF	99.1	0.023 s	16 m 52 s	19.2	74.5	0.024 s	3 m 45 s	19.1
fpn+v5	FPN	82.9	0.022 s	13 m 23 s	16.2	77.7	0.023 s	2 m 55 s	16.3
ALIEN	Hybrid	99.8	0.021 s	03 m 0 s	16.1	80.1	0.021 s	1 m 8 s	16.1

**Table 8 sensors-23-01233-t008:** Models performance evaluation II.

	Overall Model Performance (Detection and Identification)					
**Models**	**mAP**	** $R_{c}^{+}$ **	**F1**	**Time**	**Rel.**	**Space Used**
	**(%)**	**(%)**	**(%)**	**(fps)**	**Loss**	**GFLOPS**
v3	97.20	89.50	95.50	0.016s	0.0426	23.30
v5s	96.50	100.0	98.10	0.022s	0.0440	16.50
p2 + v5	98.30	100.0	99.10	0.024s	0.0587	19.10
fpn+v5	100.0	70.80	82.90	0.023s	0.0421	16.20
SqueezeNet	83.41	85.10	88.50	0.030s	0.0921	22.50
GoogleNet	89.55	88.75	89.10	0.035s	0.0685	24.10
VGG-16	89.25	83.20	86.10	0.039s	0.0781	26.50
MobileNet V2	74.50	72.05	74.01	0.476s	0.0951	18.40
ResNet	89.53	90.10	89.81	0.040s	0.0983	21.70
ALIEN	99.50	100.0	99.80	0.021s	0.0370	16.10

## Data Availability

The dataset used in this study is the VisioDect Dataset which can be accessed from the IEEE Dataport using the https://ieee-dataport.org/documents/visiodect-dataset-aerial-dataset-scenario-based-multi-drone-detection-and-identification (accessed on 20 December 2022).

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
