# Peer review of "ALIEN: Assisted Learning Invasive Encroachment Neutralization for Secured Drone Transportation Systemâ€"

_sensors, 2023, doi:10.3390/s23031233_

Round 1
Reviewer 1 Report
There are many research methods for the secured drone transportation system, and the research has been relatively mature at present. This paper mainly does some work on the artificial intelligence collaborative surveillance strategy, which has practical application value. Here are some suggestions:
1.The third paragraph of the first section of this article is too absolute. As stated in this article: The majority of neutralization techniques are either manual, militaristic, or reactive. But the article [5] does not clearly explain this point. This is a wrong reference.
2.There is an error in the last sentence of the third paragraph of Section 3.8 of this paper. 'Table ??' is a description of the error and needs to be corrected.
3.The objective function of this paper should be described before describing the proposed algorithm to clarify the purpose of this research. It is necessary to quantify the research purpose of this paper and use mathematical formulas to clarify the goal of this paper. At the same time, when describing the proposed algorithm, the purpose of this study should be specifically expressed in the text.
4.The experiments are insufficient. The authors should add more experiments to demonstrate the performance of their method. Maybe they can use Montecarlo simulations.
5. There are too many contents in sections 3 and 4 of this paper, and many of them can be omitted. The authors should make the pictures and words complement each other, instead of using a large number of words to describe the existing contents in the pictures.
Reviewer 2 Report
The article is focused on developing an AI to identify and verify different drones and their packages employing deep neural networks and different sensors. From my point of view, the article is too lengthy, and it was tedious to read. However, the experimental results prove the effectiveness of ALIEN.
Major issues:
- Specify the novelty of this article concerning your other works.
- In figure 2, what do the green and dotted-red lines mean?
- From what I understand, equations (5), and (12) are functions, Could you give the explicit expression for these equations?
- If possible, a video of how ALIEN works would be very helpful
Minor issues
- Specify the reference for Shi2018 and Grossman2018
- There are many Figures and Tables with the symbol ??
- Do not use contractions and improve your English
- For a better structure, first, define the equations and then reference them
- Table 7 must be improved
- Author contributions must be updated
Round 2
Reviewer 1 Report
The author has revised the article according to the review, and I think it can be accepted in the present form.